# Dynamics and control of typhoid fever in Sheno town, Ethiopia: A comprehensive nonlinear model for transmission analysis and effective intervention strategies

**Lema Abdela Baisa, Belela Samuel Kotola** *

Department of Mathematics, Oda Bultum University, Chiro, Ethiopia

* belelasamuel@gmail.com

**Data Availability Statement:** All data are in the manuscript and/or supporting information files.

**Funding:** The author(s) received no specific funding for this work.

## Abstract

This study presents a reliable mathematical model to explain the spread of typhoid fever, covering stages of susceptibility, infection, carrying, and recovery, specifically in the Sheno town community. A detailed analysis is done to ensure the solutions are positive, stay within certain limits, and are stable for both situations where the disease is absent and where it is consistently present. The Routh-Hurwitz stability criterion has been used and applied for the purpose of stability analysis. Using the next-generation matrix, we determined the intrinsic potential for disease transmission. It showing that typhoid fever is spreading actively in Sheno town, with cases above a critical level. Our findings reveal the instability of the disease-free equilibrium point alongside the stability of the endemic equilibrium point. We identified two pivotal factors for transmission of the disease: the infectious rate, representing the speed of disease transmission, and the recruitment rate, indicating the rate at which new individuals enter the susceptible population. These parameters are indispensable for devising effective control measures. It is imperative to keep these parameters below specific thresholds to maintain a basic reproduction number favorable for disease control. Additionally, the study carefully examines how different factors affect the spread of typhoid fever, giving a detailed understanding of its dynamics. At the end, this study provides valuable insights and specific strategies for managing the disease in the Sheno town community.

## 1. Introduction

Throughout history, humanity has faced serious outbreaks of infectious diseases that have caused widespread death, fear, disruptions in trade, and political instability [1]. One of these diseases is Typhoid Fever, caused by the bacterium Salmonella Typhi (S. Typhi), which belongs to the same bacterial family as other types like Salmonella Paratyphoid A, Salmonella Paratyphi B, Salmonella Choleraesuis, and Salmonella Type 2. After infecting a person, S. Typhi multiplies within cells of various organs, re-enters the bloodstream, and spreads to organs such as the bladder, bile ducts, and lymphatic tissue in the intestines, where it replicates rapidly. The

**Competing interests:** The authors have declared that no competing interests exist.

incubation period for typhoid fever ranges from eight to fourteen days, with the disease lasting approximately four to six weeks. Early symptoms include a prolonged fever, reduced appetite, vomiting, severe headache, constipation, general discomfort, fatigue, and a dry cough. As the illness progresses, there is typically an increase in fever and abdominal pain, often followed by diarrhea [2].

Human beings are the exclusive natural hosts and reservoirs for this infection, with the spread mainly occurring through the ingestion of food or water contaminated with feces. It's worth noting that ice cream has been identified as a significant factor in transmitting typhoid fever. Previous outbreaks have been traced back to shellfish collected from polluted waters, as well as raw fruits and vegetables grown using sewage as fertilizer. Understanding these methods of transmission is essential for addressing the historical and ongoing difficulties posed by typhoid fever [2].

The use of mathematical modeling to study infectious diseases has become an essential tool for understanding, predicting, and controlling the spread of infections. Unlike chronic disease epidemiology, where the focus is less on timing, epidemic modeling places a significant emphasis on the temporal aspect. This approach aims to capture the complex dynamics of disease transmission within populations, which often include a large number of individuals. By doing so, it provides valuable insights into how diseases spread and helps develop strategies to mitigate their impact [1].

To make it simpler to understand this group of people, a common method is to divide them into different categories. For example, we might separate them into those who can get sick, those who are currently sick, and those who have recovered. Then, we use math to show how people move between these categories over time, while also looking at the overall health of everyone in the group [3–6].

Numerous researchers have utilized mathematical modeling to intricately depict and analyze complex issues like this, investigates into the fine details to better understand and solve these challenging problems. For instance the researchers in [7], investigates into the transmission dynamics of dengue, a significant vector-borne disease threatening human populations through dynamic modeling. They explored the interaction between two strains of super-infection dengue.

Additionally another authors in [8], addresses the pressing issue of the COVID-19 pandemic by presenting a deterministic six-compartment epidemiological model. The model captures the emergence and spread of two strains of COVID-19 within a community, incorporating elements of quarantine and recovery through treatment. Furthermore, this research investigates into the transmission dynamics of the COVID-19 pandemic, with a particular focus on the role of careless infective individuals in the spread of the disease. Using mathematical models, the study explores the impact of human behavior, especially transmission by asymptomatic carriers. Through both qualitative analysis and numerical simulations, the research underscores the importance of targeted interventions to curb the spread of COVID-19. By shedding light on the intricate dynamics of the disease, the findings enhance our understanding of COVID-19 transmission and provide valuable guidance for public health strategies aimed at controlling its spread.

In Ethiopia, as in many other developing countries, unraveling the true burden of salmonellosis proves to be an intricate challenge. This complexity arises due to the constrained scope of studies and the absence of well-coordinated epidemiological surveillance systems. Adding to this difficulty is the fact that cases of salmonellosis are often not reported fully, and other diseases that are more prominent may overshadow its importance, potentially reducing attention on salmonellosis [8,9].

However, similar to what is seen in developed countries, effectively handling salmonellosis requires a comprehensive strategy. This involves closely monitoring different types of Salmonella and carefully checking how well they respond to antibiotics. By focusing on these important areas, we can gain a deeper understanding of salmonellosis and develop more precise and effective ways to control it [9].

Multiple studies conducted in Ethiopia have unveiled a concerning uptrend in antibiotic resistance among Salmonella strains, particularly to commonly utilized antimicrobials in both public health and veterinary domains. However, with the exception of S. Typhi, the identification of bacteria often extends solely to the serogroup level, and in some cases, merely to the genus level. This limitation results in Paratyphoid B and C isolates being grouped together with other members of serogroup B and C, rendering it impossible to differentiate between typhoidal and non-typhoidal salmonellae. To the best of our knowledge, it is quite remarkable that only one published study has utilized serotyping to differentiate the various serovars of Salmonella circulating within Ethiopia. This highlights the urgent need for improved surveillance and characterization efforts to gain a deeper understanding and more effectively combat the evolving landscape of Salmonella infections in the region [10].

## 2. Baseline model formulation

In the realm of addressing challenges affecting human populations, various models and methodologies have been employed. Notably, previous studies by researchers [11–14] have utilized diverse approaches. In our investigation, we have expanded upon the dynamics of typhoid fever transmission and control, leveraging and extending the model framework originally presented in reference [1]. Our extended model incorporates novel assumptions, notably the concept that a successfully treated individual may become susceptible to infection once again. In essence, those who have recovered from typhoid fever can experience a re-infection, and this transition is quantified by the rate 'k,' where individuals from the recovered group re-enter the susceptible class. Additionally, our model accounts for the influx of newborn individuals into the susceptible group, capturing this demographic change through the net birth rate 'n'. These detailed considerations help us understand better how typhoid fever spreads, how people recover from it, and how likely they are to get it again.

### 2.1 Basic assumption

In our exploration of the dynamics surrounding the spread and control of typhoid fever, we introduce nuanced assumptions that add depth and intricacy to modified modeling framework. Specifically, we acknowledge the potential for individuals who have successfully recovered from typhoid fever to undergo a subsequent infection. This special aspect of the model acknowledges that someone who has recovered from typhoid and been treated before could still get the disease again. This intriguing aspect is encapsulated by the parameter 'k,' denoting the rate at which individual transition from the recovered class back into the susceptible category.

Moreover, the model takes into account the introduction of new individuals into the susceptible group, capturing the dynamics of population growth. This increase is because new babies are born into the community, which is measured by the net birth rate $n$. By including both people getting sick again after recovering and the ongoing arrival of susceptible individuals through childbirth, our model becomes more detailed. This helps us better understand how typhoid fever spreads, taking into account both disease patterns and population changes.

**Table 1. The description of the variables and parameters of the model.**

| Variables and Parameter | Description |
|---|---|
| S | Susceptible |
| I | Infectious |
| C | Carrier |
| R | Recovered |
| $n$ | Recruitment rate in the population |
| $\mu$ | Natural death rate |
| $\alpha$ | The rate of infectious |
| $\sigma$ | The typhoid fever–indicated mortality rate |
| $\beta$ | The rate of progression from infective to carrier |
| $\delta$ | The carrier-induced mortality rate |
| $\gamma$ | The rate of recovery from the carrier stage |
| $b$ | The rate of recovery from infectious stage |
| $k$ | Re-susceptibility rate or the rate of recovered population being susceptible again |

The state variables and the parameters we have used for formulation of the model are described in Table 1 below.

The flow diagram of the model is given in Fig 1 below.

The corresponding mathematical model equations are:

$$\left.\begin{array}{l} \dfrac{dS}{dt} = n - \alpha SI - \mu S + kR, \\[2mm] \dfrac{dI}{dt} = \alpha SI - (\mu + \sigma)I - \beta I - bI, \\[2mm] \dfrac{dC}{dt} = \beta I - (\mu + \delta)C - \gamma C, \\[2mm] \dfrac{dR}{dt} = bI + \gamma C - \mu R - kR. \end{array}\right\} \tag{1}$$

with initial condition $S(0) = S_0 > 0$, $I(0) = I_0 \geq 0$, $C(0) = C_0 \geq 0$ and $R(0) = R_0 \geq 0$.

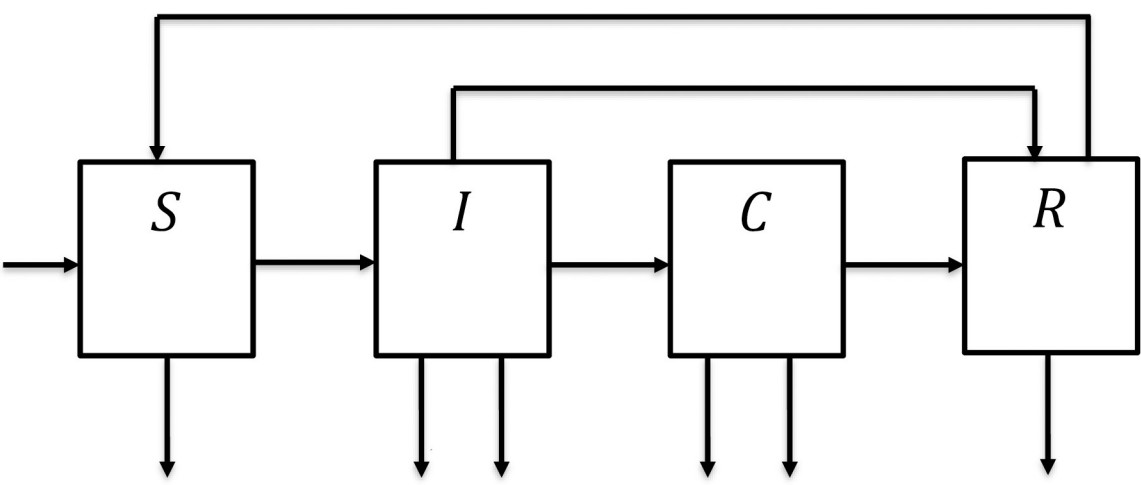

**Fig 1. Flow diagram of the model.**

## 2.2 Positivity of the solutions

To ensure that a dynamical system model is epidemiologically meaningful, it is crucial to demonstrate that all solutions with non-negative initial conditions remain non-negative. To validate this, we have examined each differential equation in the given dynamical system and proved that their solutions maintain non-negativity throughout, as detailed below.

Let us take the first differential equation

$$\frac{dS}{dt} = n - \alpha SI - \mu S + kR.$$

Then by using, the integrating factor $e^{\int_0^\tau (\alpha I + \mu) dt}$ we obtained the solution;

$$S(t) = e^{-\int_0^\tau (\alpha I + \mu) dt} \left[ \int_0^\tau (n + kR) e^{\int_0^\tau (\alpha I + \mu) dt} dt + S(0) \right],$$

$$\Rightarrow S(t) = S_0 e^{-\int_0^\tau (\alpha I + \mu) dt} + \left[ (n + kR) e^{-\int_0^\tau (\alpha I + \mu) dt} \right] \int_0^\tau e^{\int_0^\tau (\alpha I + \mu) dt} dt.$$

Since the initial susceptible $s_0$ population is also positive and the range of exponential function is positive. This implies that $S(t)$ is positive.

Applying identical procedures, it is observed that the solution for all state variables, derived from the model's formulation, preserves positivity.

## 2.3 Boundedness of the solution

To show the boundedness of the solution, we have to show lower bound and upper bound.

As a result, $N(0) = N_0 > 0$, $S(0) = S_0 > 0$, $I(0) = I_0 \geq 0$, $C(0) = C_0$ and $R(0) = R_0 \geq 0$.

Those initial conditions considered being lower bound. Now we are going to show upper bound. By taking the relation $N(t) = S(t) + I(t) + C(t) + R(t)$ and differentiating both sides of the equation with respect to time, we get $\frac{dN}{dt} = \frac{dS}{dt} + \frac{dI}{dt} + \frac{dC}{dt} + \frac{dR}{dt}$.

$$\Rightarrow \frac{dN}{dt} = (n - \alpha SI - \mu S + kR) + (\alpha SI - (\mu + \sigma)I - \beta I - bI) + (\beta I - (\mu + \delta)C - \gamma C) + (bI + \gamma C - \mu R - kR).$$

$$\Rightarrow \frac{dN}{dt} = n - N\mu - \sigma I - \delta c, \text{ since } N = S + I + C + R.$$

$$\Rightarrow \frac{dN}{dt} + N\mu = n - \sigma I - \delta c.$$

Since, we are assumed that the typhoid- induced death rate and carrier-induced death rate are equal,

That is $\delta = \sigma$

$$\Rightarrow \frac{dN}{dt} + N\mu = n - \sigma(I + C).$$

$$\Rightarrow \frac{dN}{dt} + (\mu + \sigma)N = n + \sigma(S + R).$$

This is the form of first order linear ordinary differential equation.

$$N(t) = N_0 e^{-\int_0^\tau (\mu+\sigma)dt} + \left[(n + \sigma(S+R)e^{-\int_0^\tau (\mu+\sigma)dt}\right] \int_0^\tau e^{\int_0^\tau (\mu+\sigma)dt} dt.$$

Since, any exponential function with positive coefficients is always positive, $N(t)$ is positive. That is the solution, $N(t)$ is bounded above by

$$N_0 e^{-\int_0^\tau (\mu+\sigma)dt} + [(n + \sigma(S+R)]e^{-\int_0^\tau (\mu+\sigma)dt} \int_0^\tau e^{\int_0^\tau (\mu+\sigma)dt} dt.$$

Hence, $N_0 \leq N \leq N_0 e^{-\int_0^\tau (\mu+\sigma)dt} + [(n + \sigma(S+R)]e^{-\int_0^\tau (\mu+\sigma)dt} \int_0^\tau e^{\int_0^\tau (\mu+\sigma)dt} dt.$

### 2.4 Equilibrium points

**2.4.1 The Disease Free Equilibrium Point (DFEP).** To ascertain the disease-free equilibrium point of the model, we meticulously establish equilibrium by setting the system (1) to zero. Through this meticulous process, we derive the disease-free equilibrium point, given by $(S^0, I^0, C^0, R^0) = \left(\frac{n}{\mu}, 0, 0, 0\right)$.

**2.4.2 Endemic Equilibrium Point (EEP).** To determine endemic equilibrium point, equating the Eq (1) equal to zero, where $I \neq 0$. That is $\frac{dS}{dt} = 0, \frac{dI}{dt} = 0, \frac{dC}{dt} = 0$ and $\frac{dR}{dt} = 0$.

$$\Rightarrow S = \frac{\mu + \sigma + \beta + b}{\alpha}.$$

$$\Rightarrow I = \left(\frac{\mu + \delta + \gamma}{\beta}\right)C.$$

$$\Rightarrow R = \left(\frac{b(\mu + \delta + \gamma) + \beta\gamma}{\beta(\mu + k)}\right)C.$$

$$\Rightarrow C = \frac{\beta(\mu + k)[\mu(\mu + \sigma + \beta + b) - \alpha n]}{\alpha[bk(\mu + \delta + \gamma) + k\beta\gamma - (\mu + k)(\mu + \sigma + \beta + b)(+\delta + \gamma)]}.$$

$$\Rightarrow I = \frac{(\mu + k)(\mu + \delta + \gamma)[\mu(\mu + \sigma + \beta + b) - \alpha n]}{\alpha[(bk(\mu + \delta + \gamma) + k\beta\gamma) - (\mu + k)(\mu + \sigma + \beta + b)(+\delta + \gamma)]}.$$

$$\Rightarrow R = \frac{(b(\mu + \delta + \gamma) + \beta\gamma)[\mu(\mu + \sigma + \beta + b) - \alpha n]}{\alpha[(bk(\mu + \delta + \gamma) + k\beta\gamma) - (\mu + k)(\mu + \sigma + \beta + b)(+\delta + \gamma)]}.$$

Therefore, the endemic equilibrium point $(S^*, I^*, C^*, R^*)$ is given by

$$(S^*, I^*, C^*, R^*) =$$
$$\left(\left(\frac{\mu + \sigma + \beta + b}{\alpha}\right), \left(\frac{(\mu + k)(\mu + \delta + \gamma)}{\alpha(\mu + \sigma + \beta + b)(\mu + \delta + \gamma)}\right), \left(\frac{\beta(\mu + k)(\mu(\mu + \sigma + \beta + b) - \alpha n N)}{\alpha(\mu + \delta + \gamma) + \beta\gamma)(\mu + \delta + \gamma)}\right),\right.$$
$$\left.\left(\frac{(b(\mu + \delta + \gamma) + \beta\gamma)(\mu(\mu + \sigma + \beta + b) - \alpha n N)}{\alpha(k - (\mu + k)(\mu + \sigma + \beta + b)(\mu + \delta + \gamma))}\right)\right).$$

## 2.5 Stability analysis of equilibrium points for the model

To determine the stability analysis of equilibrium point we use by Jacobian matrix.

Let's consider the differential equations as the following such that:

$$\left.\begin{array}{l} \dfrac{dS}{dt} = n - \alpha SI - \mu S + kR = f_1(S, I, C, R) \\[2mm] \dfrac{dI}{dt} = \alpha SI - (\mu + \sigma)I - \beta I - bI = f_2(S, I, C, R) \\[2mm] \dfrac{dC}{dt} = \beta I - (\mu + \delta)C - \gamma C = f_3(S, I, C, R) \\[2mm] \dfrac{dR}{dt} = bI + \gamma C - \mu R - kR = f_4(S, I, C, R) \end{array}\right\} \quad (2)$$

The Jacobian matrix of the dynamical system (2) at any equilibrium point $(S, I, C, R)$ is given by,

$$J(S, I, C, R) = \begin{bmatrix} \dfrac{\partial f_1}{\partial S} & \dfrac{\partial f_1}{\partial I} & \dfrac{\partial f_1}{\partial C} & \dfrac{\partial f_1}{\partial R} \\[2mm] \dfrac{\partial f_2}{\partial S} & \dfrac{\partial f_2}{\partial I} & \dfrac{\partial f_2}{\partial C} & \dfrac{\partial f_2}{\partial R} \\[2mm] \dfrac{\partial f_3}{\partial S} & \dfrac{\partial f_3}{\partial I} & \dfrac{\partial f_3}{\partial C} & \dfrac{\partial f_3}{\partial R} \\[2mm] \dfrac{\partial f_4}{\partial S} & \dfrac{\partial f_4}{\partial I} & \dfrac{\partial f_4}{\partial R} & \dfrac{\partial f_4}{\partial R} \end{bmatrix}$$
$$= \begin{bmatrix} -\alpha I - \mu & -\alpha S & 0 & k \\ \alpha I & \alpha S - (\mu + \sigma + \beta + b) & 0 & 0 \\ 0 & \beta & -(\mu + \delta + \gamma) & 0 \\ 0 & b & \gamma & -(\mu + k) \end{bmatrix}$$

**2.5.1 Stability analysis of disease free equilibrium point.** The Jacobian matrix at disease free equilibrium points $(S^0, I^0, C^0, R^0)$ is given as the following;

$$J\left(\frac{n}{\mu}, 0, 0, 0\right) = \begin{bmatrix} -\mu & \dfrac{-\alpha n}{\mu} & 0 & k \\[2mm] 0 & \dfrac{\alpha n}{\mu} - (\mu + \sigma + \beta + b) & 0 & 0 \\[2mm] 0 & \beta & -(\mu + \delta + \gamma) & 0 \\[2mm] 0 & b & \gamma & -(\mu + k) \end{bmatrix}$$

The characteristics equation is given by

$$a_4\lambda^4 + a_3\lambda^3 + a_2\lambda^2 + a_1\lambda + a_0 = 0.$$

Where,

$a_4 = 1,$

$a_3 = 4\mu + \sigma + \beta + b + \delta + \gamma + k - \dfrac{\alpha n}{\mu},$

$a_2 = \mu(2\mu + \delta + \gamma + k) + (\mu + k)(\mu + \delta + \gamma) - \left(\dfrac{\alpha n}{\mu} - (\mu + \sigma + \beta + b)\right)(3\mu + \delta + \gamma + k),$

$a_1 = \mu(\mu + k)(\mu + \delta + \gamma) - \left(\dfrac{\alpha n}{\mu} - (\mu + \sigma + \beta + b)\right)(\mu(2\mu + \delta + \gamma + k) + (\mu + k)(\mu + \delta + \gamma)),$

$a_0 = -\mu(\mu + k)(\mu + \delta + \gamma)\left(\dfrac{\alpha n}{\mu} - (\mu + \sigma + \beta + b)\right).$

For the characteristics equation, $a_4\lambda^4 + a_3\lambda^3 + a_2\lambda^2 + a_1\lambda + a_0 = 0$, the Routh-Hurwitz array given by:

$$
\begin{array}{c|ccc}
\lambda^4 & 1 & a_2 & a_0 \\
\lambda^3 & a_3 & a_1 & 0 \\
\lambda^2 & b_1 & b_2 & 0 \\
\lambda^1 & c_1 & c_2 & 0 \\
\lambda^0 & d_1 & 0 & 0
\end{array}
$$

Where,

$b_1 = \dfrac{-1}{a_{n-1}}\begin{vmatrix} a_n & a_{n-2} \\ a_{n-1} & a_{n-3} \end{vmatrix} = \dfrac{a_{n-1}a_{n-2} - a_n a_{n-3}}{a_{n-1}}$ here we do have $n = 4$ and $a_{n-5} =$

$a_{n-6} = \cdots = 0.$

$$b_1 = \dfrac{a_2 a_3 - a_1 a_4}{a_3} = a_2 - \dfrac{a_1 a_4}{a_3}.$$

$$b_1 = \dfrac{\left[\mu(\mu + k)(\mu + \delta + \gamma) - \left(\frac{\alpha n}{\mu} - (\mu + \sigma + \beta + b)\right)\begin{pmatrix} \mu(2\mu + \delta + \gamma + k) + \\ (\mu + k)(\mu + \delta + \gamma) \end{pmatrix}\right]}{4\mu + \sigma + \beta + b + \delta + \gamma + k - \frac{\alpha n}{\mu}} = X^*$$

$$b_2 = \dfrac{-1}{a_{n-1}}\begin{vmatrix} a_n & a_{n-4} \\ a_{n-1} & a_{n-5} \end{vmatrix} = \dfrac{a_{n-1}a_{n-4} - a_n a_{n-5}}{a_{n-1}} = \dfrac{a_3 a_0 - a_4(0)}{a_3} = \dfrac{a_3 a_0}{a_3} = a_0.$$

$$\Rightarrow b_2 = a_0 = -\mu(\mu + k)(\mu + \delta + \gamma)\left(\dfrac{\alpha n}{\mu} - (\mu + \sigma + \beta + b)\right).$$

$$b_3 = \frac{-1}{a_{n-1}} \begin{vmatrix} a_n & a_{n-6} \\ a_{n-1} & a_{n-7} \end{vmatrix} = \frac{a_{n-1}a_{n-6} - a_n a_{n-7}}{a_{n-1}} = \frac{a_3(0) - a_4(0)}{a_3} = 0.$$

$$c_1 = \frac{-1}{b_1} \begin{vmatrix} a_{n-1} & a_{n-3} \\ b_1 & b_2 \end{vmatrix} = \frac{b_1 a_{n-3} - a_{n-1}b_2}{b_1} = \frac{b_1 a_1 - a_3 b_2}{b_1} = a_1 - \frac{a_3 b_2}{b_1}.$$

$$c_1 = a_1 - \frac{a_3 b_2}{b_1}.$$

$$c_1 = \mu(\mu+k)(\mu+\delta+\gamma) - \left(\frac{\alpha n}{\mu} - (\mu+\sigma+\beta+b)\right)(\mu(2\mu+\delta+\gamma+k) + (\mu+k)(\mu+\delta+\gamma))$$

$$- \frac{\left[4\mu+\sigma+\beta+b+\delta+\gamma+k - \frac{\alpha n}{\mu}\right]\left[\begin{array}{c} -\mu(\mu+k)(\mu+\delta+\gamma) \\ \left(\frac{\alpha n}{\mu} - (\mu+\sigma+\beta+b)\right) \end{array}\right]}{X^*}.$$

$$c_2 = \frac{-1}{b_1} \begin{vmatrix} a_{n-1} & a_{n-5} \\ b_1 & b_3 \end{vmatrix} = \frac{b_1 a_{n-5} - a_{n-1}b_3}{b_1} = \frac{b_1(0) - a_3(0)}{b_1} = 0.$$

$$d_1 = \frac{-1}{c_1} \begin{vmatrix} b_1 & b_2 \\ c_1 & c_2 \end{vmatrix} = \frac{c_1 b_2 - c_2 b_1}{c_1} = \frac{c_1 b_2 - (0)b_1}{c_1} = \frac{c_1 b_2}{c_1} = b_2.$$

$$d_1 = b_2 = -\mu(\mu+k)(\mu+\delta+\gamma)\left(\frac{\alpha n}{\mu} - (\mu+\sigma+\beta+b)\right).$$

Thus based on the Routh-Hurwitz stability criterion, the disease free equilibrium point become stable if all the values of the coefficients $a_i's$ obtained above are non-zero with the same sign and if all of the first column elements of the Routh-Hurwitz array such as $a_3$, $b_1$, $c_1$ and $d_1$ given above are all positive, but if at least one of the values of $a_3$, $b_1$, $c_1$ and $d_1$ is negative, the equilibrium point becomes unstable, this is because $a_4 > 0$.

**2.5.2 Stability analysis of endemic equilibrium.** The Jacobian matrix at any point (S, I, C, R) is:

$$J(S, I, C, R) = \begin{bmatrix} -\alpha I - \mu & -\alpha S & 0 & k \\ \alpha I & \alpha S - (\mu+\sigma+\beta+b) & 0 & 0 \\ 0 & \beta & -(\mu+\delta+\gamma) & 0 \\ 0 & b & \gamma & -(\mu+k) \end{bmatrix}$$

At the endemic equilibrium point the Jacobian matrix becomes

$$J(S^*, I^*, C^*, R^*) = \begin{bmatrix} -\alpha I^* - \mu & -\alpha S^* & 0 & k \\ \alpha I^* & \alpha S^* - (\mu + \sigma + \beta + b) & 0 & 0 \\ 0 & \beta & -(\mu + \delta + \gamma) & 0 \\ 0 & b & \gamma & -(\mu + k) \end{bmatrix}$$

Where $S^* = \frac{\mu + \sigma + \beta + b}{\alpha}$ and $\alpha S^* = \alpha\left(\frac{\mu + \sigma + \beta + b}{\alpha}\right) = \mu + \sigma + \beta + b$

Hence the Jacobian matrix $J$ becomes

$$J(S^*, I^*, C^*, R^*) = \begin{bmatrix} -\alpha I^* - \mu & -\alpha S^* & 0 & k \\ \alpha I^* & 0 & 0 & 0 \\ 0 & \beta & -(\mu + \delta + \gamma) & 0 \\ 0 & b & \gamma & -(\mu + k) \end{bmatrix}$$

Therefore, the characteristics equation becomes;

$$\Rightarrow \lambda^4 + [3\mu + \delta + \gamma + k + \alpha I^*]\lambda^3 + [(\alpha I^* + \mu)(\mu + k) + (\mu + \delta + \gamma)(\mu + \delta + \gamma) +$$
$$(\alpha I^* + \mu)(\mu + \delta + \gamma) + \alpha(\mu + \sigma + \beta + b)I^*]\lambda^2 + [(\alpha I^* + \mu)(\mu + k)(\mu + \delta + \gamma) +$$
$$\alpha(\mu + \sigma + \beta + b)(2\mu + \delta + \gamma + k)I^* - \alpha k I^*(\mu + \delta + \gamma)^2]$$
$$\lambda + [\alpha(\mu + k)(\mu + \delta + \gamma)(\mu + \sigma + \beta + b)I^* - \alpha\beta\gamma k I^* - \alpha k I^*(\mu + \delta + \gamma)^3].$$

The characteristics equation can be written in the form of

$$a_4\lambda^4 + a_3\lambda^3 + a_2\lambda^2 + a_1\lambda + a_0 = 0.$$

Where,

$$a_4 = 1,$$

$$a_3 = 3\mu + \delta + \gamma + k + \alpha I^*,$$

$$a_2 = (\alpha I^* + \mu)(\mu + k) + (\mu + \delta + \gamma)(\mu + \delta + \gamma) + (\alpha I^* + \mu)(\mu + \delta + \gamma) + \alpha(\mu + \sigma + \beta + b)I^*,$$

$$a_1 = (\alpha I^* + \mu)(\mu + k)(\mu + \delta + \gamma) + \alpha(\mu + \sigma + \beta + b)(2\mu + \delta + \gamma + k)I^* - \alpha k I^*(\mu + \delta + \gamma)^2,$$

$$a_0 = \alpha(\mu + k)(\mu + \delta + \gamma)(\mu + \sigma + \beta + b)I^* - \alpha\beta\gamma k I^* - \alpha k I^*(\mu + \delta + \gamma)^3,$$

$$S^* = \frac{\mu + \sigma + \beta + b}{\alpha},$$

$$I^* = \frac{(\mu + k)(\mu + \delta + \gamma)[\mu(\mu + \sigma + \beta + b) - \alpha n]}{\alpha[k[b(\mu + \delta + \gamma) + \beta\gamma] - (\mu + k)(\mu + \sigma + \beta + b)(\mu + \delta + \gamma)]}.$$

For the characteristics equation, $\lambda^4 + a_3\lambda^3 + a_2\lambda^2 + a_1\lambda + a_0 = 0$, the Routh-Hurwitz array given by:

$$
\begin{array}{c|ccc}
\lambda^4 & 1 & a_2 & a_0 \\
\lambda^3 & a_3 & a_1 & 0 \\
\lambda^2 & b_1 & b_2 & 0 \\
\lambda^1 & c_1 & c_2 & 0 \\
\lambda^0 & d_1 & 0 & 0
\end{array}
$$

Where,

$$
b_1 = \frac{-1}{a_{n-1}} \begin{vmatrix} a_n & a_{n-2} \\ a_{n-1} & a_{n-3} \end{vmatrix} = \frac{a_{n-1}a_{n-2} - a_n a_{n-3}}{a_{n-1}} \text{ Since } n = 4, \text{ we do have}
$$

$$
b_1 = \frac{a_2 a_3 - a_1 a_4}{a_3} = a_2 - \frac{a_1 a_4}{a_3}.
$$

$$
\begin{aligned}
b_1 &= (\alpha I^* + \mu)(\mu + k) + (\mu + \delta + \gamma)(\mu + \delta + \gamma) + (\alpha I^* + \mu)(\mu + \delta + \gamma) \\
&\quad + \alpha(\mu + \sigma + \beta + b)I^* \\
&= Y^*.
\end{aligned}
$$

$$
b_2 = \frac{-1}{a_{n-1}} \begin{vmatrix} a_n & a_{n-4} \\ a_{n-1} & a_{n-5} \end{vmatrix} = \frac{a_{n-1}a_{n-4} - a_n a_{n-5}}{a_{n-1}} = \frac{a_3 a_0 - a_4(0)}{a_3} = a_0.
$$

$$
b_2 = a_0 = \alpha(\mu + k)(\mu + \delta + \gamma)(\mu + \sigma + \beta + b)I^* - \alpha\beta\gamma kI^* - \alpha kI^*(\mu + \delta + \gamma)^3.
$$

$$
b_3 = \frac{-1}{a_{n-1}} \begin{vmatrix} a_n & a_{n-6} \\ a_{n-1} & a_{n-7} \end{vmatrix} = \frac{a_{n-1}a_{n-6} - a_n a_{n-7}}{a_{n-1}} = \frac{a_3(0) - a_4(0)}{a_{n-1}} = 0.
$$

$$
c_1 = \frac{-1}{b_1} \begin{vmatrix} a_{n-1} & a_{n-3} \\ b_1 & b_2 \end{vmatrix} = \frac{b_1 a_{n-3} - a_{n-1}b_2}{b_1} = \frac{b_1 a_1 - a_3 b_2}{b_1} = a_{1-}\frac{a_3 b_2}{b_1}.
$$

$$
c_1 = (\alpha I^* + \mu)(\mu + k)(\mu + \delta + \gamma) + \alpha(\mu + \sigma + \beta + b)(2\mu + \delta + \gamma + k)I^*.
$$

Where,

$$
X^* = b_1
$$

$$
c_2 = \frac{-1}{b_1} \begin{vmatrix} a_{n-1} & a_{n-5} \\ b_1 & b_3 \end{vmatrix} = \frac{b_1 a_{n-5} - a_{n-1}b_3}{b_1} = \frac{b_1(0) - a_3 b_3}{b_1} = \frac{a_3 b_3}{b_1} = 0.
$$

$$c_2 = 0.$$

$$d_1 = \frac{-1}{c_1} \begin{vmatrix} b_1 & b_2 \\ c_1 & c_2 \end{vmatrix} = \frac{c_1 b_2 - c_2 b_1}{c_1} = b_2 - \frac{c_2 b_1}{c_1} = b_2 - \frac{(0)b_1}{c_1} = b_2.$$

$$d_1 = b_2 = a_0 = \alpha(\mu + k)(\mu + \delta + \gamma)(\mu + \sigma + \beta + b)I^* - \alpha\beta\gamma k I^* - \alpha k I^*(\mu + \delta + \gamma)^3.$$

Therefore, by the Routh stability criterion if the values of all the coefficients $a_3$, $a_2$, $a_1$ and $a_0$ obtained above are non-zero with the same algebraic sign and if all the first column elements $a_3$, $b_1$, $c_1$ and $d_1$ are positive, then the endemic equilibrium point becomes stable otherwise it is unstable.

## 2.6 Reproduction number of the model

The reproduction number, a crucial epidemiological metric indicating which is the average number of secondary infections generated by a single infected individual in a susceptible population, is determined through the next generation method. This method hinges on identifying the reproduction number as the principal eigenvalue of the next generation matrix $G = FV^{-1}$, where in F signifies new infections and $V$ denotes the transfer of infections from one compartment to another [6–8]. To ascertain the reproduction number $R_0$ using the next generation method, the differential equations governing the dynamical system are reconfigured as follows:

$$\frac{dI}{dt} = \alpha SI - (\mu + \sigma)I - \beta I - bI,$$

$$\frac{dC}{dt} = \beta I - (\mu + \delta)C - \gamma C,$$

$$\frac{dS}{dt} = n - \alpha SI - \mu S + kR,$$

$$\frac{dR}{dt} = bI + \gamma C - \mu R - kR.$$

The basic matrix of the second generation number is:

$$f(I, C, S, R) = \begin{pmatrix} \alpha SI \\ 0 \\ 0 \\ 0 \end{pmatrix} \text{ and } v(I, C, S, R) = \begin{pmatrix} (\mu + \sigma + \beta + b)I \\ (\mu + \delta + \gamma)C - \beta I \\ \mu S - (n + kR) \\ (\mu + k)R - (bI + \gamma C) \end{pmatrix}.$$

$$F = \begin{pmatrix} \alpha S^0 & 0 \\ 0 & 0 \end{pmatrix}, \text{ since } S^0 = \frac{n}{\mu}.$$

$$F = \begin{pmatrix} \alpha\dfrac{n}{\mu} & 0 \\ 0 & 0 \end{pmatrix}.$$

$$V = \begin{pmatrix} (\mu + \sigma + \beta + b) & 0 \\ -\beta & (\mu + \delta + \gamma) \end{pmatrix}.$$

$$V = \begin{pmatrix} (\mu + \sigma + \beta + b) & 0 \\ -\beta & (\mu + \delta + \gamma) \end{pmatrix}.$$

$$\Rightarrow V^{-1} = \begin{pmatrix} \frac{1}{(\mu+\sigma+\beta+b)} & 0 \\ \frac{\beta}{(\mu+\sigma+\beta+b)(\mu+\delta+\gamma)} & \frac{1}{(\mu+\delta+\gamma)} \end{pmatrix}.$$

$$\Rightarrow FV^{-1} = \begin{pmatrix} \frac{n\alpha}{\mu(\mu+\sigma+\beta+b)} & 0 \\ 0 & 0 \end{pmatrix}.$$

The corresponding characteristics polynomial equation is then given by

$$\begin{vmatrix} \frac{n\alpha}{\mu(\mu+\sigma+\beta+b)} - \lambda & 0 \\ 0 & 0 - \lambda \end{vmatrix} = 0.$$

Since reproduction number is calculated as the dominant eigenvalue of the next generation matrix, which describes the transmission dynamics of an infectious disease within a population, which is given by $\rho\left(FV^{-1}\right) = \frac{\alpha n}{\mu(\mu+\sigma+\beta+b)}$.

Therefore, $R_0 = \frac{\alpha n}{\mu(\mu+\sigma+\beta+b)}$.

## 3. Real parameter estimation

To explore the dynamics of typhoid fever transmission in Sheno town, we employ the SICRS compartmental model, delineating the total population N into four distinct categories: S (Susceptible), I (Infectious), C (Carrier), and R (Recovered). This modeling approach allows us to examine the disease's spread within the community.

In parameterizing our model, we make several key assumptions:

The disease propagates within a closed environment, implying that Sheno town's population size remains constant at N.

1. The population exhibits heterogeneous mixing, with no constraints based on age, sex, occupation, or religion.

2. Transmission occurs through direct contact or ingestion of contaminated food or liquids by susceptible individuals.

3. Newborns enter the susceptible group, transitioning from the newborn state to the susceptible group at a net birth rate denoted as n.

4. Natural mortality rates are uniform across all compartments.

5. Previously treated individuals can become reinfected.

6. The mortality rates attributed to typhoid fever and carrier-induced complications are equal.

7. We estimate the model parameters using data collected from Sheno town spanning from 01/02/2009 to 05/07/2010 E.C, supplemented by additional sources to refine our

**Table 2. The gathered data about the total population and typhoid fever in sheno town.**

| Description | Values |
| --- | --- |
| Number of total population | 15891 |
| Number of initial infected population | 1421 |
| Number of initial carriers population | 381 |
| Number of initial recovered population | 1369 |
| Number of new born in Sheno town | 1413 |
| Number of people who again become susceptible after recovered | 1040 |
| Total number of natural death in Sheno town | 451 |

parameterization. This comprehensive approach ensures the robustness and accuracy of our model's predictions.

The summary in the Table 2 provides a comprehensive overview of the collected data concerning both the total population of Sheno town and the prevalence of typhoid fever within the community. It serves as a valuable resource for understanding the demographic composition of the town and the epidemiological characteristics of the disease. Through this data, researchers can analyze trends, patterns, and correlations to gain insights into the dynamics of typhoid fever transmission and its impact on the population of Sheno town.

Table 3 presents the initial distribution of individuals within each compartment of the SICRS model utilized to study typhoid fever transmission dynamics. It provides a snapshot of the population's composition at the beginning of the simulation, distinguishing between Susceptible (S), Infectious (I), Carrier (C), and Recovered (R) categories. By detailing the initial population counts in each compartment, researchers can establish the baseline for their simulations and track changes over time to assess the model's predictive accuracy and effectiveness in representing real-world scenarios.

According to [1], the per capita recovery rate of infective individuals (b) is calculated as the inverse of the time span between the onset of typhoid fever and recovery due to treatment. This duration typically ranges from 4 to 6 weeks. For our calculations, we will assume an average treatment period of 5 weeks, or 35 days. Consequently, the recovery rate of infective individuals is determined as follows:

$$b = \frac{1}{35} = 0.0286$$

The recovery rate of carriers $\gamma$ is the inverse of time between carrier state and recovery by continuous treatment. According to [3] it is 6 weeks and beyond implying that $\gamma < 1$ per year. Taken the treatment to be 42 a days. Therefore recovery rate if infective becomes;

$$\gamma = \frac{1}{42} = 0.0238$$

**Table 3. The total number of initial populations in each of the compartments of the model.**

| Classes of the initial populations | Symbol | Total number |
| --- | --- | --- |
| Susceptible | $S_0$ | 14470 |
| Infected | $I_0$ | 1421 |
| Carrier | $C_0$ | 381 |
| Recovered | $R_0$ | 1369 |

Per capita progression rate from- infective to carrier $\beta$: This range from month to decades. Most infective never progress towards the carrier state. However, average progression is short and become shorter due to the availability of vaccines now.it is about three to five percent (3–5%) of typhoid fever patients become carriers. Thus the average range of the progression become, $\frac{(30+3650)}{2} days = 1840$ days. Therefore the progression rate becomes;

$$\beta = \frac{1}{1840} = 0.0005,$$

$$n = \frac{\text{Number of new born individual}}{\text{Total number of population}} = \frac{1413}{15891} = 0.0889,$$

$$\mu = \frac{\text{Number of total natural death}}{\text{Total population}} = \frac{451}{15891} = 0.0284,$$

$$\alpha = \frac{\text{Number of typhoid fever infectious}}{\text{Total population}} = \frac{1421}{15891} = 0.0894,$$

$$k = \frac{\text{Number of people who again into the susecptible class}}{\text{number of the recoverde people}} = \frac{1040}{1369} = 0.7597.$$

In Table 4:- we have provided a summary of the estimated and calculated values of various parameters which has been used in the model.

Since the reproduction number is given by,

$$R_0 = \frac{\alpha n}{\mu(\mu + \sigma + \beta + b)},$$

where $\alpha = 0.0894$, $n = 0.0889$, $\mu = 0.0284$, $\sigma = 0.0006$, $\beta = 0.0005$, $b = 0.0286$.

$$\Rightarrow R_0 = \frac{0.0079}{0.0016} = 4.9375.$$

## 4. Sensitivity analysis

Through the process of sensitivity indices analysis of the effective reproduction number concerning all pertinent parameters, we elucidate the criticality of each parameter in both the

**Table 4. Parameters values.**

| Descriptions | Symbols | Values | Source |
|---|---|---|---|
| The recruitment rate | $n$ | 0.0889 | Estimated |
| The natural death rate | $\mu$ | 0.0284 | Estimated |
| The rate of infection | $\alpha$ | 0.0894 | Estimated |
| The rate of recovery from the infectious stage | $b$ | 0.0286 | [3] |
| The typhoid fever-induced death rate | $\sigma$ | 0.0006 | [4] |
| The rate of progression from infective to carrier | $\beta$ | 0.0005 | [3] |
| The carrier-induced death rate | $\delta$ | 0.0006 | [4] |
| The rate of recovery from carrier stage | $\gamma$ | 0.0238 | [3] |
| The rate of re-susceptible | $k$ | 0.7597 | Estimated |

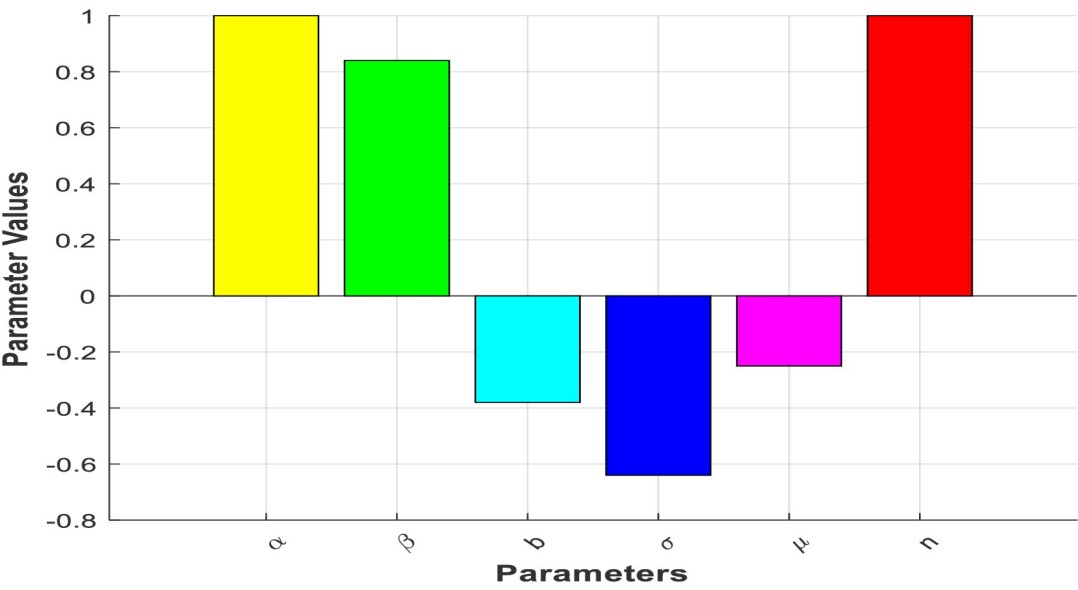

**Fig 2. Sensitivity analysis.**

spread and control of the typhoid fever model within this section. This method enables us to discern the degree of influence exerted by each parameter on the propagation of the disease, thereby facilitating targeted and efficient intervention strategies [8–10]. The sensitivity analysis prioritizes parameters based on their impact, allowing for resource allocation and control measures to be focused on those parameters with the greatest effect. Moreover, this approach enhances our understanding of disease transmission dynamics and aids in the optimization of interventions by identifying parameters that significantly affect the reproduction number. By providing a quantitative basis for public health decisions and policy formulations, sensitivity analysis supports informed decision-making in disease control efforts. In the presented analysis, we investigate the sensitivity of $R_0$ through a graphical representation of the sensitivity indices. Notably, the parameter with the highest sensitivity index demonstrates a substantial impact on $R_0$, surpassing the influence of all other parameters, as depicted in the accompanying figure.

The parameter values from sensitivity analysis of Fig 2 provide further insights into the dynamics of typhoid fever in Sheno Town, Ethiopia. The rate of infection ($\alpha$) and recruitment rate ($n$), both set to 1, indicate a high potential for disease transmission and population growth. The negative values for natural death rate ($\mu$), typhoid fever-induced death rate ($\sigma$), and the rate of recovery from the infectious stage ($b$) suggest potential decreases in overall mortality, typhoid-related mortality, and recovery from the infectious stage, respectively.

The positive value for the rate of progression from infective to carrier ($\beta$) at 0.84 suggests an increase in the transition from infective to carrier stage, possibly influencing disease persistence. These adjustments highlight the importance of understanding the interplay between different parameters in shaping the dynamics of typhoid fever.

In this section, we conducted a local sensitivity analysis of the effective reproduction number ($R_0$) concerning relevant parameters in the typhoid fever model. This involves varying each parameter individually while keeping others fixed at baseline values to observe their immediate effect on $R_0$.

This approach highlights the most influential parameters, aiding in targeted intervention strategies. Unlike global sensitivity analysis, which explores the entire parameter space, our method focuses on the linear response near specific baseline conditions. The results show that the parameter with the highest sensitivity index has a significantly greater impact on $R_0$ than all others, as illustrated in the accompanying figure.

In interpreting these results, it becomes crucial to reevaluate intervention strategies. Strategies aimed at reducing the rate of infection, enhancing recovery from the infectious stage, and addressing the progression to carrier stage may play a pivotal role in controlling typhoid fever in Sheno Town. Moreover, a comprehensive approach considering the impact on mortality rates and the potential for population growth is essential for effective public health planning and implementation. Regular monitoring and adaptability of strategies based on ongoing evaluations will be essential for dynamic and successful disease management.

## 5. Numerical simulation

In the context of the SICRS model for typhoid fever, the concept of local asymptotic stability at the disease-free equilibrium point holds significant importance. This stability indicates that, under specific conditions, the system tends to converge towards a state where no individuals within the population are infected. The pictures in this section show this important idea in a visual way. Each part of the picture represents a different part of the model, showing clearly how the system moves to or from a specific point. These visuals help us understand how the model behaves, which can guide decisions about how to control diseases better.

The picture shown as Fig 3 above is a strong way to understand how the system behaves when there's no disease. It visually shows how the system changes over time and how different parts of it interact. This graph helps us see clearly the stability of the mathematical model it represents. By watching how each group changes over time, you can see the complex patterns that determine whether the situation moves closer to or further from being disease-free. These clear visuals help us understand if the model is stable and make it easier to find important

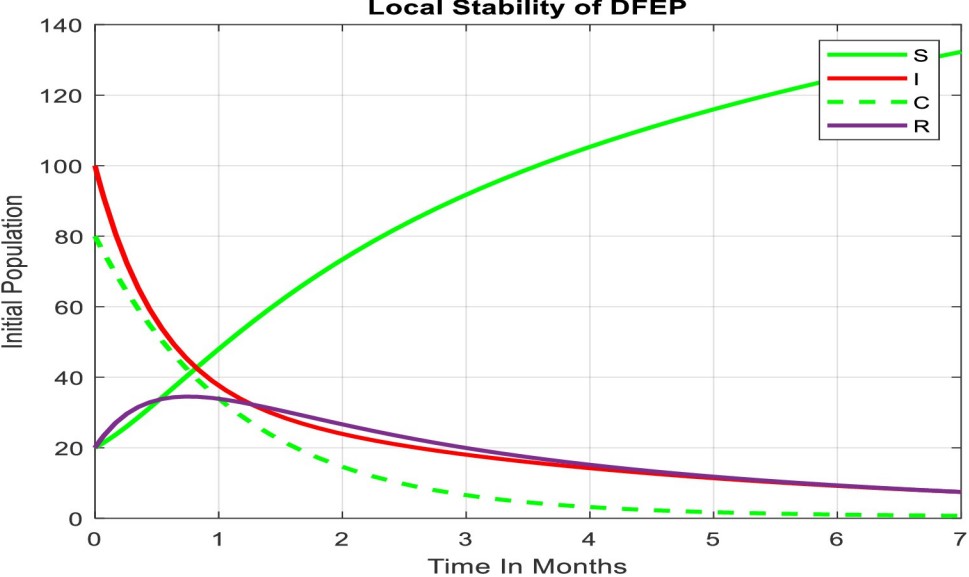

**Fig 3. Local stability of DFE.**

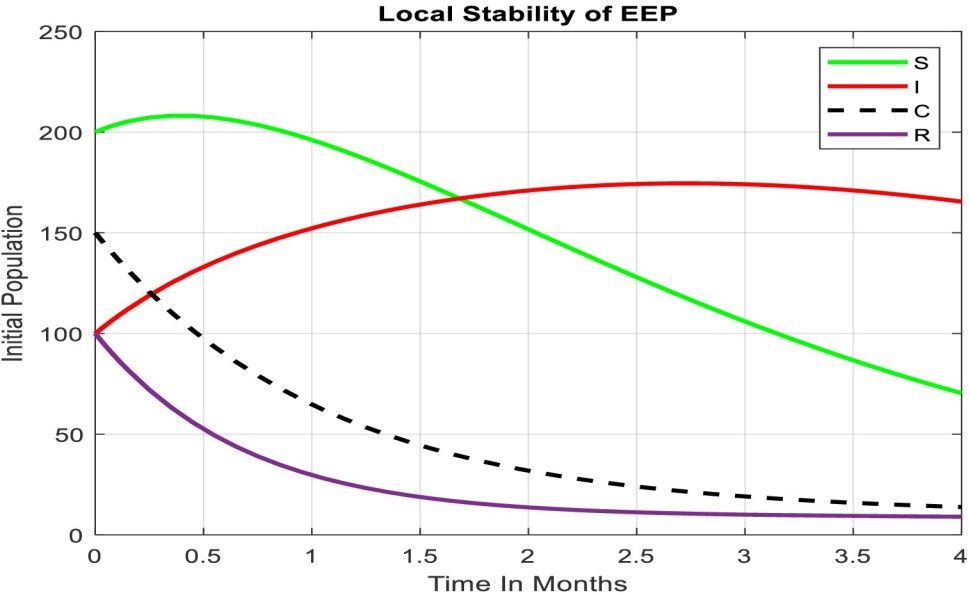

**Fig 4. Local stability of EEP.**

moments when things might become unstable [10,15]. This visual way of showing information is essential for making decisions about how to intervene and explaining why diseases spread the way they do.

Understanding why a disease stays in a population for a long time is important [16,17]. The graph in Fig 4, along with analyzing eigenvalues, helps us deeply study how the disease behaves over time. By looking at the system around its stable points, we can see if typhoid fever might continue or if it can be controlled under specific conditions in the math model. This approach helps us not only understand how diseases persist but also make plans to reduce their impact and spread. So, graphs and eigenvalue analysis are crucial for figuring out the complex dynamics and possible outcomes of controlling diseases in a modeled population.

Moreover, we consider the reproduction number of the extended SICRS mathematical model, $R_0 = \frac{\alpha n}{\mu(\mu+\sigma+\beta+b)}$, which depends on six parameters: namely the rate of infection($\alpha$), recruitment rate($n$), natural death rate($\mu$), typhoid induced death rate($\sigma$), the rate progression from infective to carrier($\beta$), the rate of recovery from the infectious stage($b$). In this analysis we discuss the effect of each parameters change on the reproduction number graphically using win plot software, where the parameters values are taken from Table 4, that is $\alpha = 0.0894$, $n = 0.0889$, $\mu = 0.0284$, $\sigma = 0.0006$, $\beta = 0.0005$, $b = 0.0286$.

1. Let we take our control parameter $\alpha$, the rate of infection and the remaining parameters are constant. That is, take $n = 0.0889$, $\mu = 0.0284$, $\sigma = 0.0006$, $\beta = 0.0005$, $b = 0.0286$.

Therefore we do have $R_0 = \left(\frac{n}{\mu(\mu+\sigma+\beta+b)}\right)\alpha$.

$$\Rightarrow R_0 = \left(\frac{0.0889}{0.0284(0.0284 + 0.0006 + 0.0005 + 0.0286)}\right)\alpha = \left(\frac{(0.0889)}{0.0284(0.0581)}\right)\alpha.$$

$$\Rightarrow R_0 = 55.5625\alpha.$$

We observe from the graph in Fig 5, that the reproduction number and the rate of infection are proportional.

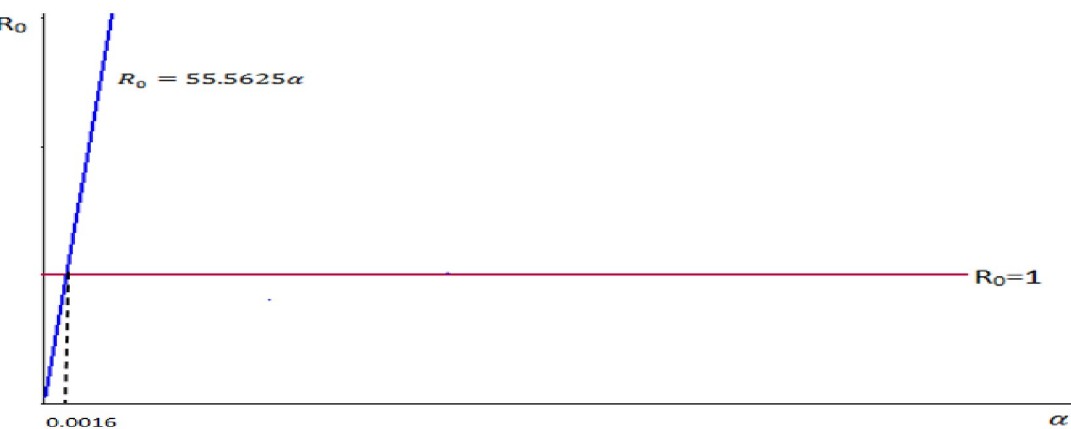

**Fig 5. Reproduction number versus the rate of infection.**

2. Let we take our control parameter $n$, recruitment rate and the remaining parameters are constant. That is, take $\alpha = 0.0894$, $\mu = 0.0284$, $\sigma = 0.0006$, $\beta = 0.0005$, $b = 0.0286$.

Therefore we do have $R_0 = \left(\frac{\alpha}{\mu(\mu+\sigma+\beta+b)}\right)n$.

$$\Rightarrow R_0 = \left(\frac{0.0894}{0.0284(0.0284 + 0.0006 + 0.0005 + 0.0286)}\right)n = \left(\frac{0.0894}{0.0284(0.0581)}\right)n.$$

$$\Rightarrow R_0 = 55.875n.$$

We observe from the Fig 6 that the reproduction number and the recruitment rate are proportional.

3. Let we take our control parameter $\mu$, natural death rate and the remaining parameters are constant. That is, take $n = 0.0889$, $\alpha = 0.0894$, $\sigma = 0.0006$, $\beta = 0.0005$, $b = 0.0286$.

Therefore $R_0 = \frac{\alpha n}{\mu(\mu+\sigma+\beta+b)} = \frac{(0.0894)(0.0889)}{\mu(\mu+0.0006+0.0005+0.0286)} = \frac{0.0079}{\mu(\mu+0.0297)} = \frac{0.0079}{\mu(\mu+0.0297)}.$

We observe from the Fig 7 that the reproduction number and the natural death rate are inversely proportional.

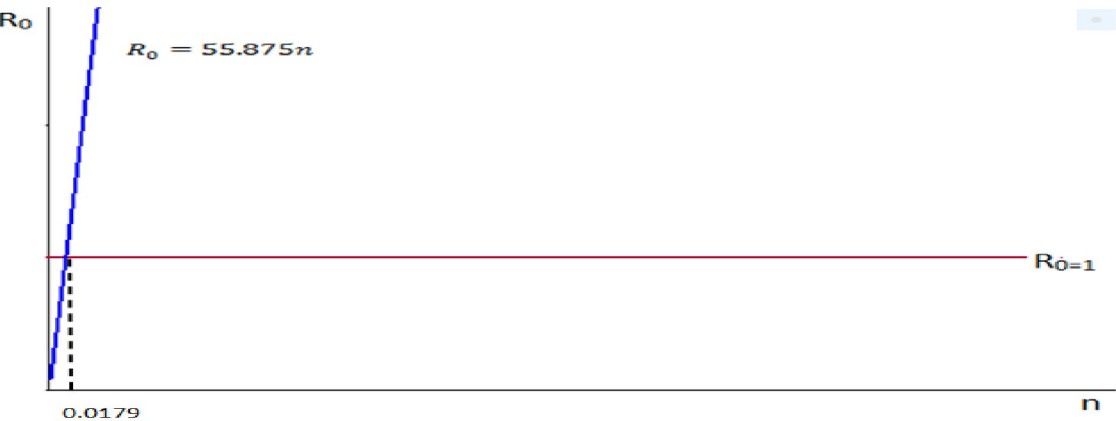

**Fig 6. Reproduction number versus recruitment rate.**

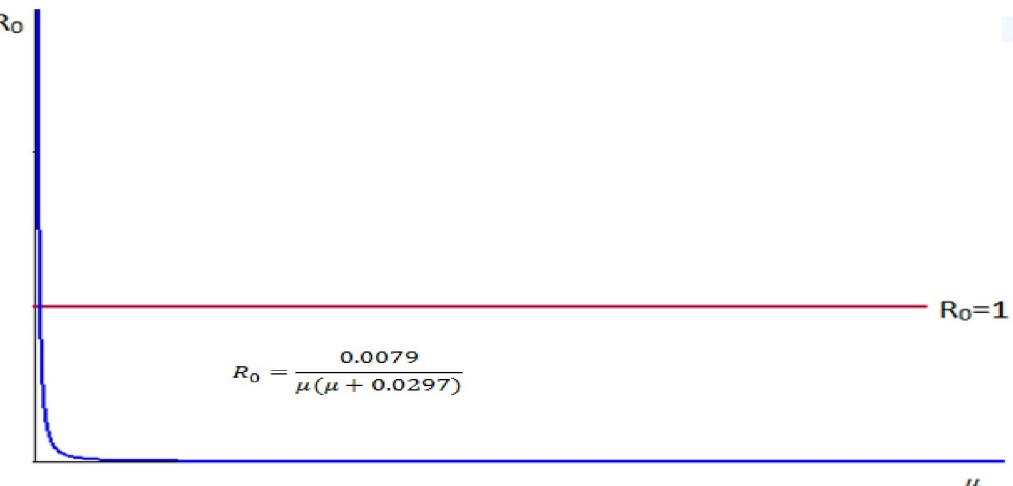

**Fig 7. Reproduction number versus natural death rate.**

4. Let we take our control parameter $\sigma$, the typhoid induced death rate and the remaining parameters are constant. That is, take $\alpha = 0.0894$, $n = 0.0889$, $\mu = 0.0284$ $\beta = 0.0005$, $b = 0.0286$. Therefore we do have $R_0 = \frac{(0.0894)(0.0889)}{0.0284(0.0284+\sigma+0.0005+0.0286)} = \frac{0.0079}{0.0284(\sigma+0.0575)}$.

$$\Rightarrow R_0 = \frac{0.0079}{0.0284\sigma + 0.0016}.$$

We observe from Fig 8 that the reproduction number and the typhoid induced death rate are inversely proportional.

5. Let we take our control parameter $\beta$, the rate of progression from infective to carrier and the remaining parameters are constant. That is, take $\alpha = 0.0894$, $n = 0.0889$, $\mu = 0.0284$, $\sigma = 0.0006$, $b = 0.0286$.

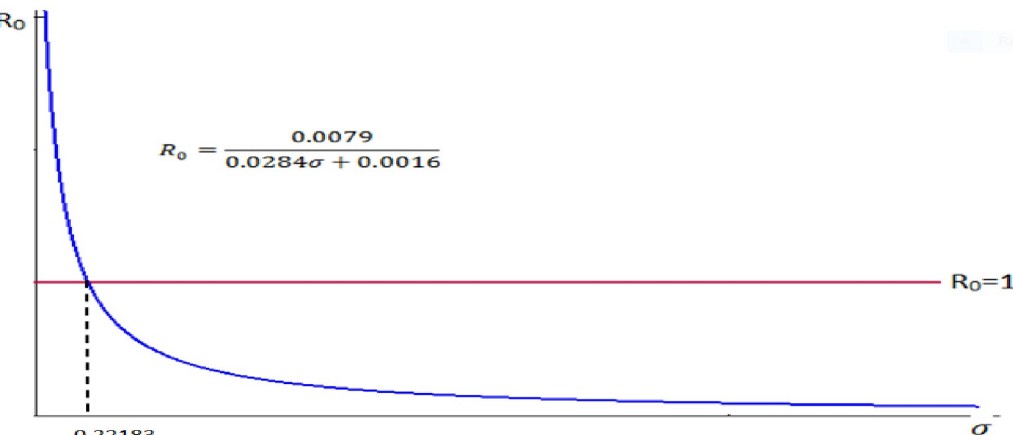

**Fig 8. Reproduction number versus typhoid induced death rate.**

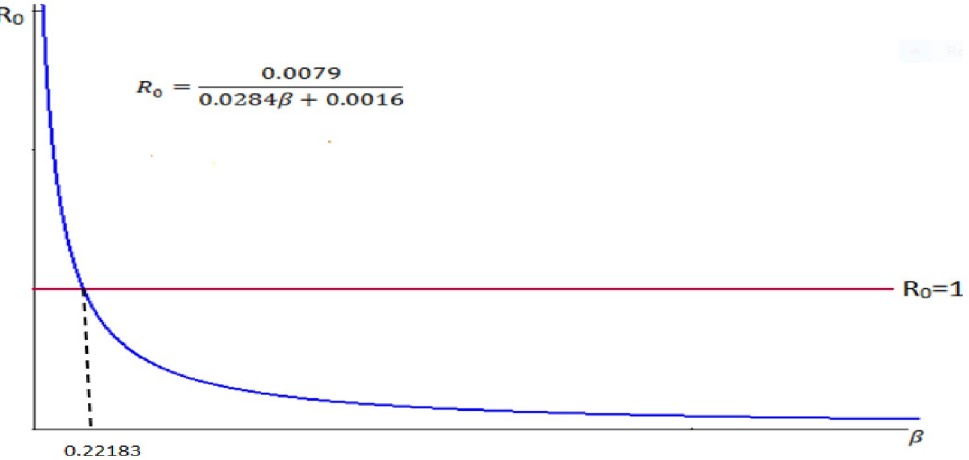

**Fig 9. Reproduction number versus rate of progression from infective to carrier.**

Therefore we do have $R_0 = \frac{(0.0894)(0.0889)}{0.0284(0.0284+0.0006+\beta+0.0286)} = \frac{0.0079}{0.0284(\beta+0.0576)}$.

$$\Rightarrow R_0 = \frac{0.0079}{0.0284\beta + 0.0016}.$$

We observe from the Fig 9 that the reproduction number and the rate of progression from infective to carrier are inversely proportional.

6. Let we take our control parameter $b$, the rate of recovery from infectious stage and the remaining parameters are constant.

That is, take $\alpha = 0.0894$, $n = 0.0889$, $\mu = 0.0284$, $\sigma = 0.0006$, $\beta = 0.0005$.

Therefore we do have $R_0 = \frac{(0.0894)(0.0889)}{0.0284(0.0284+0.0006+0.0005+b)} = \frac{0.0079}{0.0284(b+0.0295)} = \frac{0.0079}{0.0284b+0.0008}$.

We observe from Fig 10 that the reproduction number and the rate of recovery from infectious stage are inversely proportional.

## 6. Result and discussion

The SICRS model, designed to study typhoid fever dynamics using a deterministic system of differential equations, proves to be a robust tool for predicting and comprehending disease

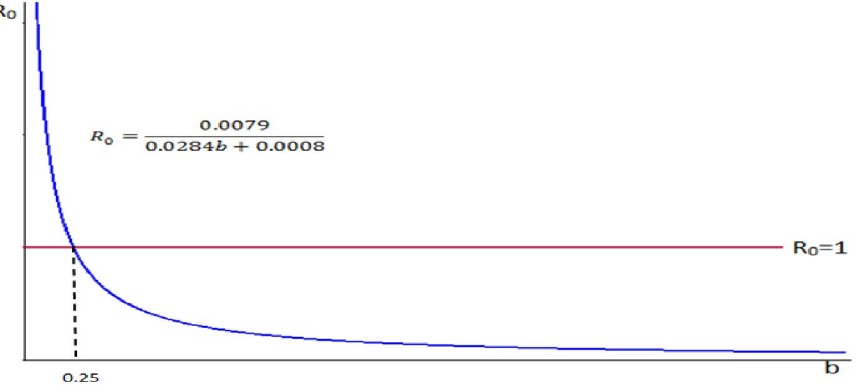

**Fig 10. Reproduction number versus rate of recovery from the infectious stage.**

transmission patterns. In this investigation, we have conducted a thorough qualitative analysis, ensuring both epidemiological and mathematical integrity by verifying the positivity and boundedness of solutions. The model unveiled two crucial equilibrium points: one representing the absence of the disease and the other signifying its persistent presence. To understand how typhoid fever might spread in Sheno town, we used a key measure in epidemiology called the basic reproduction number ($R_0$), a fundamental metric in epidemiology. Through the incorporation of actual data, our expanded model produced an $R_0$ value of 4.9375, pointing to a worrisome degree of disease transmission.

To explore what our results mean, we used computer simulations and studied how things stay balanced. The Routh-Hurwitz criteria helped us understand how stable the equilibrium points were, as shown in references [18–20], revealing the disease-free equilibrium as unstable and the endemic equilibrium as stable. Our graphical representations vividly illustrate the impact of varying parameters on $R_0$ and disease spread. Notably, increased infection rates and recruitment rates correlated with higher $R_0$ values, signifying heightened disease transmission [21,22]. Conversely, lower values of these parameters resulted in decreased $R_0$, indicating a reduction in the spread of typhoid fever.

We identified that changes in the rate of infection and recruitment significantly influenced $R_0$, demonstrating their pivotal roles in disease dynamics. Additionally, the typhoid-induced death rate, progression rate, and rate of recovery from the infective stage exhibited clear impacts on $R_0$, offering valuable insights for targeted interventions. Importantly, our results underscored the effectiveness of reducing infection and recruitment rates in controlling the transmission of typhoid fever. In general, our study combines mathematical rigor with real-world applicability to provide a nuanced understanding of typhoid fever dynamics in Sheno town.

The graphical representations not only enhance the interpretability of our results but also serve as powerful tools for communicating the implications of various interventions. This research contributes valuable insights for public health strategies, emphasizing the significance of targeted measures to curb infection and recruitment rates for effective typhoid fever control in Sheno town and similar settings.

## 7. Conclusion and recommendation

### 7.1 Conclusion

In this paper, a non-linear mathematical model was formulated and then we have seen their stability analysis. From the stability analysis and numerical simulation result, the researcher concluded that the disease spread through the community of Sheno town without control measure and the number of infective individuals increases in the community. Since the reproduction number $R_0 = 4.9375$ which is greater than one.

### 7.2 Recommendation

To reduce the spread of typhoid fever in Sheno town we recommended the following suggestion.

1. Control the parametric values of infectious rate $\alpha < 0.0016$, by fixing the remaining parametric values.

2. Control the parametric values of recruitment rate $n < 0.0179$, by fixing the remaining parametric values.

In general when these two control parametric value $\alpha < 0.0016$ and $n < 0.0179$, the SICRS model reproduction number less than one, then the transmission of disease decrease and the disease will be controlled.

Furthermore, we propose extending and refining our work by incorporating additional assumptions that consider fractional aspects, such as fractional order derivatives. This approach, as utilized by other researchers like [23,24], introduces memory effects into disease models. The inclusion of memory effects through fractional derivatives allows for a more accurate representation of phenomena where past states influence current dynamics, thereby enhancing the model's ability to capture real-world complexities and dynamics more effectively.

## 8. Limitation of the research

For the purpose of this study, we have considered only Kebeles of Sheno town without including the nearby Kebeles, which can get health service at Sheno town health institution. In addition, we considered all new born was healthy themselves and they enter in to susceptible class directly and we collected data by ignoring the immigration and migration of population into and out of Sheno town respectively. Thus, we suggest that by giving attention for this limitation and making improvements on them, the future work will be better on controlling the spread of typhoid fever.

## Author Contributions

**Conceptualization:** Lema Abdela Baisa.

**Formal analysis:** Belela Samuel Kotola.

**Investigation:** Lema Abdela Baisa.

**Methodology:** Belela Samuel Kotola.

**Software:** Belela Samuel Kotola.

**Writing – original draft:** Lema Abdela Baisa.

**Writing – review & editing:** Belela Samuel Kotola.

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
