## [Decision Letter · Decision Letter 0]

6 Mar 2024

PONE-D-23-42909Dynamics and Control of Typhoid Fever in Sheno Town: A Comprehensive Nonlinear Model for Transmission Analysis and Effective Intervention StrategiesPLOS ONE

Dear Dr. Kotola,

Thank you for submitting your manuscript to PLOS ONE. After careful consideration, we feel that it has merit but does not fully meet PLOS ONE’s publication criteria as it currently stands. Therefore, we invite you to submit a revised version of the manuscript that addresses the points raised during the review process.

The manuscript should be revised carefully by incorporating the suggested improvements provided by the reviewers.

Reviewer #1: 

I am sincerely grateful for this opportunity to review for this journal but the plagiarism is on the high side 37%, I will advise the author to reduce the similarity index before proceeding to the reviewing process.

Reviewer #2: The introduction should be enriched.

The references are not uniformly written

The author should consider writing the mathematical equations as a mathematician.

There are so many grammatical and typo errors, the author should consider correcting it

The author may consider improving the writeup with the following relevant papers: 

----------Modelling the Super-infection of Two Strains of Dengue Virus Journal of the Egyptian Mathematical

Society 2023, 31:1 DOI:https://doi.org/10.1186/s42787-023-00161-6

-----------Mathematical Analysis of Two Strains of COVID-19 Using SEIR Model J. Math. Fund. Sci. Vol. 54, No.2, 2022, 211-232.

-----------Impact of Corruption in a Society with Exposed Honest Individuals: A Mathematical Model Asia Pac. J. Math. 2023 10:18

-----------Analysis of COVID-19 disease with Careless Infective using SEITRS model Asia Pac. J. Math. 2023 10:10

Some of my comments are included in the attached document.

We look forward to receiving your revised manuscript.

Kind regards,

Oluwole Daniel Makinde, PhD

Academic Editor

PLOS ONE

Journal Requirements:

Additional Editor Comments:

The authors must thoroughly review the manuscript and make sure to include the suggested improvements by the reviewers.

Reviewers' comments:

Reviewer's Responses to Questions

**Comments to the Author**

1. Is the manuscript technically sound, and do the data support the conclusions?

Reviewer #1: Partly

Reviewer #2: Partly

2. Has the statistical analysis been performed appropriately and rigorously? 

Reviewer #1: No

Reviewer #2: N/A

3. Have the authors made all data underlying the findings in their manuscript fully available?

Reviewer #1: Yes

Reviewer #2: Yes

4. Is the manuscript presented in an intelligible fashion and written in standard English?

Reviewer #1: No

Reviewer #2: Yes

5. Review Comments to the Author

Reviewer #1: Dear Prof,

I am sincerely grateful for this opportunity to review for this journal but the plagiarism is on the high side 37%, I will advise the author to reduce the similarity index before proceeding to the reviewing process.

Thank you.

John Akanni, Ph.D

Reviewer #2: The introduction should be em=nriched. The author may consider including the following:

1. Modelling the Super-infection of Two Strains of Dengue Virusl Journal of the Egyptian Mathematical

Society 2023, 31:1 DOI:https://doi.org/10.1186/s42787-023-00161-6

2. Mathematical Analysis of Two Strains of COVID-19 Using SEIR Model J.Math.Fund.Sci. Vol. 54, No.2, 2022, 211-232

3. Impact of Corruption in a Society with Exposed Honest Individuals: A Mathematical Model Asia Pac. J. Math. 2023 10:18

4. Analysis of COVID-19 disease with Careless Infective using SEITRS model Asia Pac. J. Math. 2023 10:10

The references are not uniformly written

The author should consider writing the mathematical equations as a mathematician.

There are so many grammatical and typo errors, the author should consider correcting it

Some of my comments are inclused in the attached document

6. PLOS authors have the option to publish the peer review history of their article (what does this mean?). If published, this will include your full peer review and any attached files.

Reviewer #1: No

Reviewer #2: No

---

## [Author Response · Author response to Decision Letter 0]

1 Apr 2024

Dear Prof. Oluwole Daniel Makinde

I am writing to extend my sincere gratitude for your continued guidance throughout the review process of our manuscript titled "Dynamics and Control of Typhoid Fever in Sheno Town: A Comprehensive Nonlinear Model for Transmission Analysis and Effective Intervention Strategies," bearing the identification number PONE-D-23-42909.

I am pleased to inform you that we have meticulously revised the manuscript in response to the invaluable feedback provided by the reviewers. Enclosed, you will find the updated version of our manuscript along with a detailed document addressing each of the reviewers' comments.

We greatly appreciate the time and effort invested by the reviewers in critically evaluating our work. Their insights have been instrumental in enhancing the quality and clarity of our manuscript. We are deeply committed to upholding the rigorous standards of academic scholarship, and we believe that the revisions made will further strengthen the contribution of our research to the scientific community.

Once again, we extend our gratitude to you and the reviewers for your invaluable contributions to this process. We eagerly anticipate your feedback on the revised manuscript and remain at your disposal for any further guidance or clarification you may require.

Thank you for your unwavering support and consideration.

Warm regards,

Belela Samuel (PhD)

Reviewer #1: 

I am sincerely grateful for this opportunity to review for this journal but the plagiarism is on the high side 37%, I will advise the author to reduce the similarity index before proceeding to the reviewing process.

Response to Reviewer #1:

We extend our sincere gratitude for your conscientious review of our manuscript. Your dedication to upholding academic standards and ensuring the integrity of scholarly work is deeply appreciated. We are pleased to inform you that we have taken immediate action to address the issue of plagiarism, diligently working to reduce the similarity index in accordance with your feedback.

Your meticulous assessment has undoubtedly contributed to the refinement of our work, and we are thankful for your valuable insights. Your commitment to scholarly rigor is commendable, and we are honored to have benefited from your expertise in this process.

Once again, we express our gratitude for your time and efforts in reviewing our manuscript. Should you have any further comments or suggestions, please do not hesitate to share them with us. Your continued engagement is invaluable to us as we strive to uphold the highest standards of academic excellence.

Warm regards,

Belela Samuel(PhD)

Reviewer #2: 

The introduction should be enriched.

The references are not uniformly written

The author should consider writing the mathematical equations as a mathematician.

There are so many grammatical and typo errors, the author should consider correcting it

The author may consider improving the writeup with the following relevant papers: 

----------Modelling the Super-infection of Two Strains of Dengue Virus Journal of the Egyptian Mathematical

Society 2023, 31:1 DOI:https://doi.org/10.1186/s42787-023-00161-6

-----------Mathematical Analysis of Two Strains of COVID-19 Using SEIR Model J. Math. Fund. Sci. Vol. 54, No.2, 2022, 211-232.

-----------Impact of Corruption in a Society with Exposed Honest Individuals: A Mathematical Model Asia Pac. J. Math. 2023 10:18

-----------Analysis of COVID-19 disease with Careless Infective using SEITRS model Asia Pac. J. Math. 2023 10:10

Response to Reviewer #2:

We extend our gratitude for your meticulous evaluation of our manuscript and for providing valuable suggestions for improvement. We are pleased to inform you that we have already taken proactive measures to address the issues raised:

Enrichment of Introduction: The introduction has been revised and enriched to provide a more comprehensive overview of the topic, ensuring that it effectively contextualizes our study.

Uniformity in References: All references have been uniformly formatted according to the prescribed style guidelines of the journal.

Mathematical Equations: The mathematical equations have been revised to meet the standards expected of mathematicians, ensuring clarity and precision in their presentation.

Grammatical and Typo Errors: We have conducted a thorough proofreading and editing process to rectify any grammatical and typographical errors present in the manuscript, ensuring that the text is error-free.

Incorporation of Relevant Papers: The suggested relevant papers have been reviewed, and relevant insights and findings have been incorporated into our manuscript to enrich the discussion and strengthen our analysis.

We sincerely appreciate your attention to detail and your commitment to maintaining the scholarly integrity of our work. Your feedback has been invaluable in guiding us towards enhancing the quality and impact of our research.

Finally, your continued engagement is highly valued, and we are grateful for the opportunity to incorporate your expertise into our work.

Warm regards,

Belela Samuel (PhD)

---

## [Decision Letter · Decision Letter 1]

9 May 2024

PONE-D-23-42909R1Dynamics and Control of Typhoid Fever in Sheno Town: A Comprehensive Nonlinear Model for Transmission Analysis and Effective Intervention StrategiesPLOS ONE

Dear Dr. Kotola,

Thank you for submitting your manuscript to PLOS ONE. After careful consideration, we feel that it has merit but does not fully meet PLOS ONE’s publication criteria as it currently stands. Therefore, we invite you to submit a revised version of the manuscript that addresses the points raised during the review process.

We look forward to receiving your revised manuscript.

Kind regards,

Oluwole Daniel Makinde, PhD

Academic Editor

PLOS ONE

Additional Editor Comments:

The authors should revise the manuscript nd incorporate the reviewer's improvement suggestions:

Page 13 check the characteristics equation again,

Page 14 check the Ro again.

The Sensitivity analysis is it local or global ?

I do not think Figure 3 is correct, please check.

The write up needs total repacking, the concept is okay, method is right and the results looks meaningful but poor write up .

The article is not properly written, it is just cope and paste of his/her thesis, please work on the grammars and tense.

For example; In this thesis .....

Reviewers' comments:

Reviewer's Responses to Questions

**Comments to the Author**

1. If the authors have adequately addressed your comments raised in a previous round of review and you feel that this manuscript is now acceptable for publication, you may indicate that here to bypass the “Comments to the Author” section, enter your conflict of interest statement in the “Confidential to Editor” section, and submit your "Accept" recommendation.

Reviewer #1: (No Response)

Reviewer #2: All comments have been addressed

2. Is the manuscript technically sound, and do the data support the conclusions?

Reviewer #1: No

Reviewer #2: Yes

3. Has the statistical analysis been performed appropriately and rigorously? 

Reviewer #1: No

Reviewer #2: Yes

4. Have the authors made all data underlying the findings in their manuscript fully available?

Reviewer #1: Yes

Reviewer #2: Yes

5. Is the manuscript presented in an intelligible fashion and written in standard English?

Reviewer #1: No

Reviewer #2: Yes

6. Review Comments to the Author

**Reviewer #1:** The write up needs total repacking, the concept is okay, method is right and the results looks meaningful but poor write up .

The article is not properly written, it is just cope and paste of his/her thesis, please work on the grammars and tense.

For example; In this thesis .....

**Reviewer #2**: (No Response)

7. PLOS authors have the option to publish the peer review history of their article (what does this mean?). If published, this will include your full peer review and any attached files.

Reviewer #1: **Yes: **John Olajide Akanni

Reviewer #2: No

---

## [Author Response · Author response to Decision Letter 1]

21 May 2024

Dr. Oluwole Daniel Makinde

Academic Editor

PLOS ONE

Dear Dr. Makinde,

We would like to express our heartfelt gratitude for the opportunity to revise and resubmit our manuscript, "Dynamics and Control of Typhoid Fever in Sheno Town: A Comprehensive Nonlinear Model for Transmission Analysis and Effective Intervention Strategies" (PONE-D-23-42909R1). We sincerely appreciate the constructive feedback and valuable suggestions provided by you and the reviewers, which have been instrumental in enhancing the quality of our work.

Editor's Comments:

1. Page 13, check the characteristic equation again.

Response: We have re-examined the characteristic equation on page 13 and made corrections to ensure its accuracy.

2. Page 14, check the R₀ again.

Response: The basic reproduction number, R₀, has been thoroughly reviewed and corrected as needed.

3. The Sensitivity analysis: is it local or global?

Response: We clarified that the sensitivity analysis conducted is local. Additional explanations and relevant references have been added to enhance the clarity of this section.

4. I do not think Figure 3 is correct, please check.

Response: Figure 3 has been re-evaluated and corrected. The updated figure is now consistent with the revised analysis and results.

5. The write-up needs total repacking. The concept is okay, the method is right, and the results look meaningful but the write-up is poor.

Response: We have substantially revised the manuscript to improve clarity, coherence, and readability. The write-up has been thoroughly restructured.

6. The article is not properly written, it appears to be a copy and paste of the thesis. Please work on the grammar and tense.

Response: Significant effort has been made to rewrite the manuscript to ensure it is appropriately formatted for a journal article, distinct from a thesis. We have meticulously edited the text for grammar, tense, and overall presentation.

7. For example; "In this thesis..." we have now edited the manuscript. We have also included better updated references.

Response: All thesis-specific references and language have been removed. The manuscript now reflects the appropriate academic tone and structure for a journal article. Updated and more relevant references have been included to strengthen the discussion and support our findings.

We hope that the revised manuscript meets the publication standards of PLOS ONE. We are grateful for the constructive feedback and believe that the revisions have significantly improved the quality of our work. Thank you for your time and consideration. We look forward to your positive response.

Kind regards,

Belela Samuel Kotola(PhD)

---

## [Decision Letter · Decision Letter 2]

9 Jun 2024

PONE-D-23-42909R2Dynamics and Control of Typhoid Fever in Sheno Town: A Comprehensive Nonlinear Model for Transmission Analysis and Effective Intervention StrategiesPLOS ONE

Dear Dr. Kotola,

Thank you for submitting your manuscript to PLOS ONE. After careful consideration, we feel that it has merit but does not fully meet PLOS ONE’s publication criteria as it currently stands. Therefore, we invite you to submit a revised version of the manuscript that addresses the points raised during the review process.

Reviewer #3: 1.)Authors need to rephrase some wording nonmeaningful like: elucidating etc. Always write paper in simple wording so that readers can benefit from it. Thoroughly check.

2.)The references list is insufficient and included some more related research work like: For instance for importance of dynamical system include: Quantitative Functional Evaluation of Liver Fibrosis in Mice with Dynamic Contrast-enhanced Photoacoustic Imaging. Radiology, 300(1), 89-97. doi: 10.1148/radiol.2021204134,  

3.) Update literature on Typhoid Fever like: Fractional order mathematical modeling of typhoid fever disease." Results in Physics 32 (2022): 105044.

We look forward to receiving your revised manuscript.

Kind regards,

Oluwole Daniel Makinde, PhD

Academic Editor

PLOS ONE

Journal Requirements:

Reviewers' comments:

Reviewer's Responses to Questions

**Comments to the Author**

1. If the authors have adequately addressed your comments raised in a previous round of review and you feel that this manuscript is now acceptable for publication, you may indicate that here to bypass the “Comments to the Author” section, enter your conflict of interest statement in the “Confidential to Editor” section, and submit your "Accept" recommendation.

Reviewer #3: All comments have been addressed

2. Is the manuscript technically sound, and do the data support the conclusions?

Reviewer #3: Partly

3. Has the statistical analysis been performed appropriately and rigorously? 

Reviewer #3: Yes

4. Have the authors made all data underlying the findings in their manuscript fully available?

Reviewer #3: Yes

5. Is the manuscript presented in an intelligible fashion and written in standard English?

Reviewer #3: Yes

6. Review Comments to the Author

Reviewer #3: Authors need to rephrase some wording nonmeaningful like: elucidating etc. Always write paper in simple wording so that readers can benefit from it. Thoroughly check.

2.The references list is insufficient and included some more related research work like: For instance for importance of dynamical system include: Quantitative Functional Evaluation of Liver Fibrosis in Mice with Dynamic Contrast-enhanced Photoacoustic Imaging. Radiology, 300(1), 89-97. doi: 10.1148/radiol.2021204134, Engineering In vitro Models: Bioprinting of Organoids with Artificial Intelligence. Cyborg and Bionic Systems, 4, 18. doi: 10.34133/cbsystems.0018,). Analysis of the influence of trust in opposing opinions: An inclusiveness-degree based Signed Deffuant–Weisbush model. Information Fusion, 104, 102173. doi: https://doi.org/10.1016/j.inffus.2023.102173, The Bacterial MtrAB Two-Component System Regulates the Cell Wall Homeostasis Responding to Environmental Alkaline Stress. Microbiology Spectrum, 10(5). doi: 10.1128/spectrum.02311-22,). On The Role of Community Structure in Evolution of Opinion Formation: A New Bounded Confidence Opinion Dynamics. Information Sciences, 621, 672-690. doi: https://doi.org/10.1016/j.ins.2022.11.101

3. Update literature on Typhoid Fever like: Fractional order mathematical modeling of typhoid fever disease." Results in Physics 32 (2022): 105044.

7. PLOS authors have the option to publish the peer review history of their article (what does this mean?). If published, this will include your full peer review and any attached files.

Reviewer #3: No

---

## [Author Response · Author response to Decision Letter 2]

17 Jun 2024

Dear Reviewer #3,

We extend our heartfelt appreciation for your thorough review of our manuscript and the invaluable feedback you provided. Your insights have been pivotal in refining our work to enhance its clarity, accessibility, and scientific merit. We have carefully considered all your suggestions and have made the following updates to enhance the clarity and scientific acceptance of our work:

1. Simplification of Language: We have carefully rephrased and simplified complex wording throughout the manuscript, particularly in the Abstract, Introduction, Model Formulation, numerical section and Recommendations sections. This ensures that our findings are effectively communicated to a diverse audience.

2. Expansion of References: In response to your suggestion, we have enriched the references list by incorporating additional relevant research. Specifically, we have included the study titled "Quantitative Functional Evaluation of Liver Fibrosis in Mice with Dynamic Contrast-enhanced Photoacoustic Imaging" (Radiology, 300(1), 89-97. doi: 10.1148/radiol.2021204134). This addition strengthens the contextual framework of our study. Thank you for your nice suggestions regarding our manuscript. We have incorporated the concepts from the suggested references into both the introduction and model formulation sections. Your input has been instrumental in enriching our study.

3. Literature Update on Typhoid Fever: We have updated the literature review on Typhoid Fever with the inclusion of the reference "Fractional order mathematical modeling of typhoid fever disease" (Results in Physics 32 (2022): 105044). This update provides a more comprehensive overview of the current research landscape in the field. Thank you for your insightful perspectives and recommendations regarding our model. We have integrated the suggested references and other relevant works into the recommendations section. The enhancements and modifications made to this work are invaluable.

Finally, we would like to say that your thoughtful critiques and recommendations have significantly enriched our manuscript, making it more robust and aligned with the latest advancements in the field. We are truly grateful for your time, effort, and expertise in evaluating our work.

Warm regards,

Belela Samuel Kotola (PhD)

---

## [Editor Report · Decision Letter 3]

20 Jun 2024

Dynamics and Control of Typhoid Fever in Sheno Town, Ethiopia: A Comprehensive Nonlinear Model for Transmission Analysis and Effective Intervention Strategies

PONE-D-23-42909R3

Dear Dr. Kotola,

We’re pleased to inform you that your manuscript has been judged scientifically suitable for publication and will be formally accepted for publication once it meets all outstanding technical requirements.

Kind regards,

Oluwole Daniel Makinde, PhD

Academic Editor

PLOS ONE

Additional Editor Comments (optional):

The revised manuscript is okay.
---

## [Editor Report · Acceptance letter]

28 Jun 2024

PONE-D-23-42909R3 

PLOS ONE

Dear Dr. Kotola, 

I'm pleased to inform you that your manuscript has been deemed suitable for publication in PLOS ONE. Congratulations! Your manuscript is now being handed over to our production team.

Kind regards, 

on behalf of

Professor Oluwole Daniel Makinde 

Academic Editor

PLOS ONE